# Cardiovascular risk factors and COVID-19 outcomes in hospitalised patients: a prospective cohort study

Didier Collard [ORCID],[1] Nick S Nurmohamed [ORCID],[1,2] Yannick Kaiser [ORCID],[1] Laurens F Reeskamp [ORCID],[1] Tom Dormans,[3] Hazra Moeniralam,[4] Suat Simsek,[5] Renee Douma,[6] Annet Eerens,[7] Auke C Reidinga,[8] Paul W G Elbers,[9] Martijn Beudel,[10] Liffert Vogt [ORCID],[11] Erik S G Stroes,[1] Bert-Jan H van den Born [ORCID] [1]

► Prepublication history and additional materials for this paper is available online. To view these files, please visit the journal online (http://dx.doi.org/10.1136/bmjopen-2020-045482).

DC, NSN and YK contributed equally.

For numbered affiliations see end of article.

**Correspondence to**
Dr Bert-Jan H van den Born; b.j.vandenborn@amsterdamumc.nl

## ABSTRACT

**Objectives** Recent reports suggest a high prevalence of hypertension and diabetes in COVID-19 patients, but the role of cardiovascular disease (CVD) risk factors in the clinical course of COVID-19 is unknown. We evaluated the time-to-event relationship between hypertension, dyslipidaemia, diabetes and COVID-19 outcomes.

**Design** We analysed data from the prospective Dutch CovidPredict cohort, an ongoing prospective study of patients admitted for COVID-19 infection.

**Setting** Patients from eight participating hospitals, including two university hospitals from the CovidPredict cohort were included.

**Participants** Admitted, adult patients with a positive COVID-19 PCR or high suspicion based on CT-imaging of the thorax. Patients were followed for major outcomes during the hospitalisation. CVD risk factors were established via home medication lists and divided in antihypertensives, lipid-lowering therapy and antidiabetics.

**Primary and secondary outcomes measures** The primary outcome was mortality during the first 21 days following admission, secondary outcomes consisted of intensive care unit (ICU) admission and ICU mortality. Kaplan-Meier and Cox regression analyses were used to determine the association with CVD risk factors.

**Results** We included 1604 patients with a mean age of 66±15 of whom 60.5% were men. Antihypertensives, lipid-lowering therapy and antidiabetics were used by 45%, 34.7% and 22.1% of patients. After 21-days of follow-up; 19.2% of the patients had died or were discharged for palliative care. Cox regression analysis after adjustment for age and sex showed that the presence of ≥2 risk factors was associated with increased mortality risk (HR 1.52, 95% CI 1.15 to 2.02), but not with ICU admission. Moreover, the use of ≥2 antidiabetics and ≥2 antihypertensives was associated with mortality independent of age and sex with HRs of, respectively, 2.09 (95% CI 1.55 to 2.80) and 1.46 (95% CI 1.11 to 1.91).

**Conclusions** The accumulation of hypertension, dyslipidaemia and diabetes leads to a stepwise increased risk for short-term mortality in hospitalised COVID-19 patients independent of age and sex. Further studies investigating how these risk factors disproportionately affect COVID-19 patients are warranted.

### Strengths and limitations of this study

► While previous data reported a high prevalence of cardiovascular disease risk factors in COVID-19 patients, this study investigated whether diabetes, dyslipidaemia and hypertension predict adverse outcomes.

► This study is limited by the use of medication as surrogate for cardiovascular risk factors.

► The causality of the investigated risk factors remains to be addressed in future studies.

## INTRODUCTION

The global spread of COVID-19, first identified in Wuhan, China, in December 2019, has ignited an unprecedented ongoing global pandemic.[1] Although most infected individuals experience only mild symptoms that do not require hospitalisation, the absolute number of patients requiring hospital admission is staggering. Risk stratification of these patients is crucial to optimise the use of hospital resources.[2] Several associations with adverse outcomes in COVID-19 patients have been identified, including factors that also predispose to cardiovascular disease (CVD), such as older age, male sex, hypertension, overweight and diabetes.[3 4] Furthermore, individuals with overt CVD appear to be affected more seriously by COVID-19 infection.[5]

The association between cardiovascular events and infectious diseases is well established. Examples include the increased prevalence of myocardial infarction during influenza pandemics,[6] and the higher number of cardiac complications in patients hospitalised for community-acquired pneumonia.[7] However, there are conflicting data whether the presence of shared CVD and COVID-19 risk factors merely reflect advanced age and history of ischaemic heart disease in patients who develop severe

BMJ

infection, or are independently associated with adverse outcomes in the COVID-19 patient population.[4 8] For example, a higher than expected prevalence of diabetes, hypertension, obesity and history of CVD was reported during the previous outbreak of the Middle East respiratory syndrome coronavirus (MERS-CoV), which shares many similarities with COVID-19.[9]

In the present study, we hypothesised that three major risk independent CVD risk factors are associated with adverse outcomes in COVID-19 patients. To this end, we evaluated the time-to-event relationship between COVID-19 disease outcomes and a history of medication use for hypertension, dyslipidaemia and diabetes mellitus in a large prospective Dutch cohort of hospitalised COVID-19 patients.

## PATIENTS AND METHODS
### Study design

CovidPredict is a Dutch multicentre initiative to collect data of hospitalised patients with confirmed COVID-19.[10] For this study, patients from eight participating hospitals, including two university hospitals were included. All hospitalised patients >18 years with a positive COVID-19 PCR or high suspicion based on CT-imaging of the thorax were included (CO-RADS [COVID-19 Reporting and Data System] score 4 or 5).[11 12] A waiver for the use of hospital record data was obtained from the Medical Ethical Committees of the participating centres. Patients were given the opportunity to opt out.

The collected data were updated with daily reports on vital signs, laboratory results, complications and clinical outcomes. In addition, the use of antihypertensive, lipid-lowering and/or antidiabetic medication was determined from the home medication list. These were used as surrogates for hypertension, dyslipidaemia and diabetes. Antihypertensive medication was categorised as using either 0, 1, more than 1 of the following categories: non-dihydropyridine calcium channel blockers, renin–angiotensin system (RAS) inhibitors (either angiotensin receptor antagonist or angiotensin receptor blockers) and diuretics (either loop diuretics, thiazide or thiazide like diuretics or potassium channel blockers). Lipid-lowering therapy was classified as the use of statin, ezetimibe, fibrates or proprotein convertase subtilisin/kexin type 9 (PCSK9) inhibitors. Antidiabetic medication was classified as using either 0, 1 or more than 1 of the following classes: metformin, sulfonylurea derivates, glucagon-like peptide-1 receptor (GLP-1) agonists, (dipeptidyl peptidase-4) DDP4 inhibitors, sodium-glucose co-transporter-2 (SGLT-2) inhibitors and insulin. Obesity was defined as a body mass index >30 kg/m$^2$. Smoking was categorised into current or non-smoker/former smoker. The combined use of beta-blockers and platelet aggregation inhibitors was used as a surrogate for history of ischaemic cardiac disease. Outcomes were determined 3 and 6 weeks after admission, or earlier when the patient died or was discharged from the hospital.

### Primary and secondary outcomes

The primary outcome consisted of overall mortality during the first 21 days following admission. Overall mortality was defined as either mortality during admission or discharge for palliative care, either at home or a palliative care facility. If the patient was discharged alive from the hospital and no further follow-up data were available, we considered the patient to be event free for the whole study period. Secondary outcomes consisted of ICU admission and mortality in the subset of patients who had been admitted to the ICU.

### Statistical analysis

For the analysis, we included all patients who were primarily admitted to one of the participating centres between 27 February 2020 and 4 July 2020. Patients with unknown medication use prior to hospitalisation were excluded. Patients were categorised based on the presence of either 0, 1 or more than 1 medication based cardiovascular risk factors (hypertension, dyslipidaemia, diabetes). Baseline characteristics were depicted as mean±SD for normally distributed data, non-normally as median (IQR) or as number (percentage) for categorical variables, and were compared using the appropriate tests (analysis of variance, Kruskal-Wallis, $\chi^2$). The relationship between outcomes and cumulative cardiovascular risk factors was determined using a Kaplan-Meier analysis. We then determined HRs using Cox regression after adjustment for age and sex, and with additional adjustment for obesity, smoking and history of ischaemic cardiac disease. For ICU mortality, a landmark analysis was performed starting from ICU admission. Next, with Cox regression models using the same covariates we determined the association between mortality and the use of antihypertensive, lipid-lowering and antidiabetic drugs. The proportional hazard assumptions were verified using Schoenfeld residuals. All statistical analyses were conducted with R V.3.6.3 (R Foundation, Vienna, Austria) using the Survival V.3.1–11 and Tableone V.0.11.1 packages.

### Patient and public involvement

This study was performed during the first wave of the COVID-19 pandemic in the Netherlands. Therefore, it was not possible to involve patients or the public in the design, or conduct, or reporting, or dissemination plans of our research. All CovidPredict study results will be communicated via www.covidpredict.org.

## RESULTS
### Patient characteristics

Between 27 February 2020 and 4 July 2020, a total of 1614 patients with a confirmed COVID-19 infection were primarily admitted to one of the participating centres in the Netherlands. After exclusion of patients with an unknown medication list, we included 1604 patients in the present analysis (table 1). Their mean age was 66±15 years, 67.6% were Caucasian and 60.5% were men. 6.5% of the

**Table 1** Baseline characteristics

| | Overall | 0 RF | 1 RF | ≥2 RF | P value |
|---|---|---|---|---|---|
| n | 1604 | 680 | 400 | 524 | |
| Age (mean (SD)) | 65.67 (15.06) | 58.59 (15.71) | 69.93 (13.01) | 71.62 (11.45) | <0.001 |
| Women | 633 (39.5) | 298 (43.8) | 170 (42.5) | 165 (31.5) | <0.001 |
| Chronic cardiac disease | 467 (29.2) | 71 (10.5) | 117 (29.3) | 279 (53.4) | <0.001 |
| Hypertension | 734 (46.0) | 80 (11.8) | 242 (60.8) | 412 (78.8) | <0.001 |
| Chronic pulmonary disease | 288 (18.0) | 92 (13.6) | 82 (20.6) | 114 (21.9) | <0.001 |
| Asthma | 179 (11.2) | 77 (11.4) | 41 (10.3) | 61 (11.7) | 0.793 |
| Chronic kidney disease | 150 (9.4) | 17 (2.5) | 48 (12.0) | 85 (16.3) | <0.001 |
| Diabetes | 411 (25.7) | 19 (2.8) | 66 (16.5) | 326 (62.3) | <0.001 |
| Malignant neoplasm | 98 (6.2) | 36 (5.3) | 25 (6.3) | 37 (7.1) | 0.455 |
| Chronic haematological disorder | 57 (3.6) | 32 (4.7) | 11 (2.8) | 14 (2.7) | 0.1 |
| Smoking | 78 (6.5) | 30 (6.0) | 16 (5.4) | 32 (7.8) | 0.383 |
| Obesity | 464 (30.8) | 164 (26.1) | 112 (29.4) | 188 (37.8) | <0.001 |
| Combined use of beta-blockers and antiplatelet drugs | 167 (10.4) | 7 (1.0) | 35 (8.8) | 125 (23.9) | <0.001 |
| Antihypertensive-Rx | | | | | <0.001 |
| 0 | 883 (55.0) | 680 (100.0) | 145 (36.2) | 58 (11.1) | |
| 1 | 377 (23.5) | 0 (0.0) | 141 (35.2) | 236 (45.0) | |
| 2 | 282 (17.6) | 0 (0.0) | 95 (23.8) | 187 (35.7) | |
| ≥3 | 62 (3.9) | 0 (0.0) | 19 (4.8) | 43 (8.2) | |
| Lipid-lowering-Rx | | | | | <0.001 |
| 0 | 1047 (65.3) | 0 (0.0) | 296 (74.0) | 89 (13.5) | |
| ≥1 | 557 (34.7) | 0 (0.0) | 104 (26.0) | 453 (86.5) | |
| Glucose-lowering-Rx | | | | | <0.001 |
| 0 | 1250 (77.9) | 680 (100.0) | 359 (89.8) | 211 (40.3) | |
| 1 | 170 (10.6) | 0 (0.0) | 23 (5.8) | 147 (28.1) | |
| ≥2 | 184 (11.5) | 0 (0.0) | 18 (4.5) | 166 (31.7) | |

P values indicate comparison between subgroups based on the presence of cumulative RFs.
RF, risk factor; Rx, medication.

admitted patients were current smokers. A total of 1526 patients (95.7%) had a positive PCR for COVID-19 during the hospitalisation. The majority of admitted patients (924, 57.6%) used some form of cardiovascular medication prior to admission. Antihypertensive medication was used by 721 (45.0%) patients, of whom 497 (68.9%) used RAS-inhibitors, 256 (35.5%) calcium-antagonists and 374 (51.9%) diuretics. Lipid-lowering therapy was used by 557 (34.7%) patients, predominantly consisting of statins (540, 96.9%). In total, 354 (22.1%) patients used antidiabetic medication, of whom 282 (79.7%) used metformin and 137 patients (38.7%) insulin. A total of 167 patients (10.4%) used the combination of a beta-blocker and a platelet aggregation inhibitor, reflecting a history of ischaemic cardiac disease. In the subset of 566 patients for whom a more detailed medical history was available, we found a similar prevalence of CVD, with 13.1% of the patients having a history of coronary artery disease, 3.7% of heart failure, 7.4% of stroke and 1.6% of peripheral arterial disease (see online supplemental file 1).

### Cardiovascular risk factors as markers for mortality and ICU admission

In the entire cohort, 308 (19.2%) of the patients died or were discharged for palliative care. In total 273 (17.0%) of the patients were admitted to the ICU, of whom 78 died. A total of 1100 (68.6%) patients were discharged alive from the hospital, 50 (3.1%) were transferred to another hospital. The remaining patients were still admitted to the hospital at time of the data collection (126; 7.9%) or follow-up data were not available yet (20; 1.25%). The Kaplan-Meier analysis showed a significant association between cardiovascular risk factors and overall mortality (p<0.0001; figure 1A), with a 21-day mortality, of respectively, 11.9%, 20.5% and 29.1% for patients with 0, 1 and ≥2 CVD risk factors. We found no association between

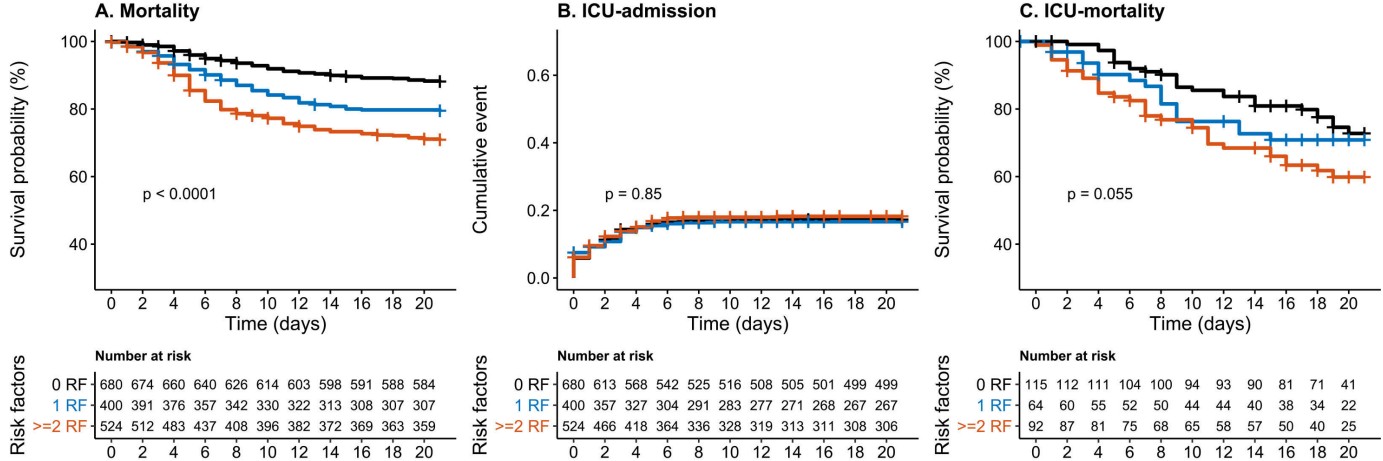

**Figure 1** Survival and time-to event analysis of cumulative cardiovascular risk factors on mortality, ICU admission and ICU mortality. Kaplan-Meier analysis of the combination of hypertension, dyslipidaemia and diabetes stratified into 0, 1 or more risk factors versus adverse clinical outcomes. (A) Depicts mortality. (B) ICU admission. (C) Depicts ICU mortality, for which a landmark analysis following ICU admission was performed. Log-rank test was used to test for differences between curves. ICU, intensive care unit; RF, risk factor.

cardiovascular risk factors and ICU admission (p=0.85; figure 1B), with an unadjusted admission rate, of respectively, 17.3%, 16.6% and 18.2%. In patients admitted to the ICU, we found a trend towards increased mortality, with an ICU mortality of 27.2%, 29.1%, 40.1% for patients with 0, 1 and ≥2 CVD-risk factors (p=0.055; figure 1C). In Cox regression analysis, a 5-year age increase was associated with an HR 1.37, 95% CI 1.31 to 1.45) for mortality, while there was no significant association with sex (HR 1.02, 95% CI 0.81 to 1.28). The presence of two or more cardiovascular risk factors was significantly associated with overall mortality (HR 1.52, 95% CI 1.15 to 2.02), but not with ICU admission or ICU mortality (table 2). After additional adjustment for smoking, obesity and the combined use of beta-blockers and platelet aggregation inhibitors the presence of two or more risk factors remained associated with mortality (HR 1.38, 95% CI 1.02 to 1.86),),

while there was no increased risk in the group with 1 CVD risk factor (HR 1.01, 95% CI 0.73 to 1.39).

### Individual risk factors
In the Cox regression models adjusted for age and sex, we observed that the use of two or more different classes of antihypertensives and antidiabetics were associated with 21-day mortality, with HR of, respectively, of 1.46 (95% CI 1.11 to 1.91) and 2.09 (95% CI 1.55 to 2.80). Similarly, we found an HR of 1.25 (95% CI 0.99 to 1.56) for the use of lipid-lowering medication. Additional adjustment for smoking, obesity and the combined use of beta-blockers and antiplatelet medication attenuated the association between the use of antihypertensive and lipid-lowering medication towards 1.33 (95% CI 1.01 to 1.76) and 1.14 (95% CI 0.89 to 1.45) for the use of ≥2 antihypertensive drugs or ≥1 lipid-lowering drug, respectively.

**Table 2** Effect of cumulative cardiovascular risk factors (RFs) on primary and secondary outcomes

| | | Model 1 | | | Model 2 | | |
|---|---|---|---|---|---|---|---|
| | Cumulative RFs | HR | 95% CI | P value | HR | 95% CI | P value |
| Mortality | 0 RF | 1 (ref) | | | 1 (ref) | | |
| | 1 RF | 1.04 | 0.76 to 1.43 | 0.786 | 1.01 | 0.73 to 1.39 | 0.956 |
| | ≥2 RF | 1.52 | 1.15 to 2.02 | 0.004 | 1.38 | 1.02 to 1.86 | 0.034 |
| ICU admission | 0 RF | 1 (ref) | | | 1 (ref) | | |
| | 1 RF | 1.11 | 0.80 to 1.53 | 0.534 | 1.06 | 0.77 to 1.47 | 0.723 |
| | ≥2 RF | 1.15 | 0.85 to 1.56 | 0.355 | 1.07 | 0.77 to 1.48 | 0.681 |
| ICU mortality | 0 RF | 1 (ref) | | | 1 (ref) | | |
| | 1 RF | 0.86 | 0.46 to 1.60 | 0.625 | 1.01 | 0.73 to 1.39 | 0.957 |
| | ≥2 RF | 1.52 | 0.90 to 2.55 | 0.115 | 1.38 | 1.02 to 1.86 | 0.035 |

Cox regression for the effect of cumulative cardiovascular risk factors on mortality, ICU admission and ICU mortality, after adjustment for sex and age (model 1), and with additional adjustments for obesity, smoking and history of coronary artery disease (model 2).
ICU, intensive care unit.

**Table 3** Effect of antihypertensive, lipid-lowering and antidiabetic medications on mortality

| Mortality | Model 1 | | | Model 2 | | |
| --- | --- | --- | --- | --- | --- | --- |
| | HR | 95% CI | P value | HR | 95% CI | P value |
| 0 antihypertensive-Rx | 1 (ref) | | | 1 (ref) | | |
| 1 antihypertensive-Rx | 1.08 | 0.81 to 1.43 | 0.597 | 1.04 | 0.78 to 1.38 | 0.790 |
| ≥2 antihypertensive-Rx | 1.46 | 1.11 to 1.91 | 0.006 | 1.33 | 1.01 to 1.76 | 0.043 |
| 0 lipid-lowering-Rx | 1 (ref) | | | 1 (ref) | | |
| ≥1 lipid-lowering-Rx | 1.25 | 0.99 to 1.56 | 0.058 | 1.14 | 0.89 to 1.45 | 0.292 |
| 0 antidiabetic-Rx | 1 (ref) | | | 1 (ref) | | |
| 1 antidiabetic-Rx | 1.34 | 0.97 to 1.85 | 0.077 | 1.25 | 0.9 to 1.74 | 0.182 |
| ≥2 antidiabetic-Rx | 2.09 | 1.55 to 2.80 | <0.001 | 1.93 | 1.43 to 2.62 | <0.001 |

Cox regression for the effect of the number of antihypertensive, lipid-lowering and antidiabetic drug classes on 21-day mortality, defined as either hospital mortality or discharge for palliative care, with adjustment for age and sex (model 1), and with additional adjustments for obesity, smoking and history of coronary artery disease (model 2).
Rx, medication.;

The association between the use of two or more glucose-lowering medications and mortality remained significant with an adjusted HR of 1.93 (95% CI 1.43 to 2.62; table 3). The HRs for the other covariates included in the regression analysis can be found in online supplemental file 2.

## DISCUSSION

In a large Dutch cohort of hospitalised COVID-19 patients, we observed that patients with more than one risk factor for CVD had a 52% higher 3-week mortality risk, independent of age and sex. In addition, our data show that the use of two or more antihypertensives or antidiabetics, or one lipid-lowering drug is associated with adverse outcomes in COVID-19 patients. Patients using two or more antidiabetic drugs had the highest mortality risk. This suggests that patients with a history of or at high risk for CVD have an increased risk for adverse COVID-19 disease outcomes.

The prevalence of medication use for CVD risk factors was higher in COVID-19 patients than in previously described cohorts and is representative for the general Dutch population of similar age.[13] Wuhan-based COVID-19 cohorts were the first to describe a higher mortality in those with hypertension and diabetes.[14] However, the average age and reported prevalence of CVD risk factors was much lower in these cohorts than in the present study.[4] The increased case-fatality rate in Europe and the USA compared with China may in part be attributable to demographic differences in the COVID-19 infected population.[15] Recent European and US-based cohorts indeed demonstrate a higher average age with a similar distribution of hypertension and diabetes compared with our study, and also show a higher mortality in COVID-19 patients with hypertension and diabetes.[16 17] In line with the results from a large US-based cohort, we observed that age was significantly associated with mortality, while we found no significant difference in mortality between sexes.[18] This suggests that although men are more often

hospitalised, there is no substantial difference in mortality after admission for severe COVID-19 infection. We add to these findings that the accumulation of CVD risk factors is associated with mortality, independent of age, sex, presence of coronary artery disease, smoking and obesity in hospitalised patients.

Analogous to COVID-19, CVD risk factors are also prevalent among patients hospitalised for community-acquired pneumonia.[19] However, we observed a stronger association between hypertension, diabetes, dyslipidaemia and mortality in COVID-19 patients compared with previous studies on community-acquired pneumonia, despite a similar prevalence of CVD risk factors.[20] In addition, these effects remained significant after adjustment for covariates such as smoking, obesity and the use of beta-blockers and antiplatelet drugs as surrogate for a history of ischaemic cardiac disease. This might suggest that CVD risk factors in COVID-19 patients disproportionally affect the clinical course of COVID-19 patients compared with other infectious diseases. The present findings are comparable to the previous MERS-CoV outbreak, which also saw a preponderance of hypertension and diabetes in hospitalised patients.[9] Because both coronaviruses enter the cell through the ACE2 receptor, a hypothesis has become that upregulation of ACE2 in patients with hypertension and/or diabetes facilitates transmission of the virus.[21–24] However, recent studies have shown no association between the use of RAS medication and the disease course of COVID-19.[25 26] Furthermore, this theory does not explain the increased risk of mortality paired with other CVD risk factors in COVID-19 patients. An alternative explanation is that CVD risk factors predispose to myocardial injury in COVID-19 infected patients, which is likely to contribute to a more severe clinical course.[27 28] Two recent studies revealed viral RNA in the myocardium of COVID-19 patients, suggesting that SARS-CoV-2 might infect the heart directly. It can be speculated that those with (preclinical) atherosclerosis are prone to experience

coronary ischaemia from viral myocardial involvement.[29 30]

In our cohort of COVID-19 patients, the presence of diabetes had the strongest association with mortality and remained significant after adjustment for covariates such as smoking, obesity and the use of beta-blockers and antiplatelet drugs as surrogate for coronary artery disease. We observed smaller effect sizes for the presence of hypertension and dyslipidaemia. These findings might suggest that diabetes predisposes for adverse outcomes in COVID-19 patients not only through its association with CVD, but potentially via other pathophysiological pathways specific to diabetes. A recent Chinese cohort showed diabetes to be associated with higher ICU admission and more in-hospital mortality, but not independently of hypertension and history of CVD.[8] Interestingly, in those with diabetes, hypertension was associated with in-hospital death, independently of history of CVD, further supporting the additive effect of CVD risk factors on COVID-19 mortality. Nevertheless, it is hard to disentangle the precise relation based on epidemiological data. In line with current guidelines on CVD risk management, cardiovascular medication from different classes were often prescribed together in our cohort, also in patients with diabetes.[31] This makes it difficult to assess their separate contribution to mortality. It remains, therefore, unknown whether diabetes alone is associated with a higher risk of adverse outcomes or whether it is merely a reflection of increased vascular ageing in combination with the other risk factors.[32] In contrast to our results, the recent cohort of Cummings *et al* found that among cardiovascular risk factors, only chronic cardiac disease was a strong predictor for hospital mortality, while smaller associations were found for other risk factors, including diabetes.[18] In line, a recent Italian cohort of hospitalised COVID-19 patients showed an increased prevalence of hypertension and diabetes among non-survivors, but only diabetes was an independent predictor after adjustment for other comorbidities.[33] In the present study, the cumulative presence of CVD risk factors did not show an association with increased risk for ICU admission. This may have been influenced by selection prior to ICU admission, where the presence of comorbidity was taken into account in the shared decision-making process, leading to a relative under-representation of patients with CVD risk factors.

The present analysis has several limitations. First, data collection was based on data collection forms of the WHO, which did not include detailed information on CVD history. For this reason, we relied on medication use as a surrogate marker for established cardiovascular risk factors or disease, which has been used before in big cohort studies.[34] Nevertheless, some of these drugs might have been prescribed for different indications, The current study was not designed to assess the relationship between specific drugs and COVID-19 outcomes. Second, we only obtained follow-up during the first 21 days, however, as depicted in the Kaplan-Meier analysis, almost all events occurred during the first 14 days, in line with earlier descriptions.[4] Finally, we cannot exclude that mortality in the current study is partially caused by other factors than COVID-19. However, as we used 21-day mortality as our primary outcome and only included patients admitted to the hospital with confirmed COVID-19 infection, it is very likely that the majority of deaths were directly attributable to COVID-19.

In conclusion, the accumulation of CVD risk factors leads to a stepwise increased risk for short-term mortality in hospitalised COVID-19 patients. Patients with diabetes had the highest risk, followed by similar risks for hypertension and dyslipidaemia. Mechanistic studies investigating how CVD risk factors disproportionately affect COVID-19 patients compared with other infectious diseases are warranted.

**Author affiliations**
[1]Department of Vascular Medicine, Amsterdam University Medical Centers, University of Amsterdam, Amsterdam, The Netherlands
[2]Department of Cardiology, Amsterdam University Medical Centers, Vrije Universiteit Amsterdam, Amsterdam, The Netherlands
[3]Department of Intensive Care, Zuyderland Medical Centre Sittard-Geleen, Sittard-Geleen, The Netherlands
[4]Internal Medicine, Sint Antonius Hospital, Nieuwegein, The Netherlands
[5]Department of Internal Medicine, Noordwest Ziekenhuisgroep, Alkmaar, The Netherlands
[6]Internal Medicine, Flevoziekenhuis, Almere, Flevoland, The Netherlands
[7]Oncology, Treant Healthcare Group, Amsterdam, The Netherlands
[8]Intensive Care, Martini Ziekenhuis, Groningen, Groningen, The Netherlands
[9]Department of Intensive Care, Amsterdam University Medical Centers, Vrije Universiteit Amsterdam, Amsterdam, The Netherlands
[10]Department of Neurology, Amsterdam University Medical Centers, University of Amsterdam, Amsterdam, The Netherlands
[11]Department of Nephrology, Amsterdam University Medical Centers, University of Amsterdam, Amsterdam, The Netherlands

**Acknowledgements** We would like to thank the CovidPredict consortium (www. covidpredict.org) for their efforts in providing the patient data.

**Contributors** DC, NSN, YK, LFR, ESGS, B-JHvdB, LV, MB and PWGE conceptualised and designed the study. TD, HM, SS, RD, AE and ACR contributed substantially to the acquisition of data. DC, NSN, YK and LFR performed the data analysis. DC, NSN, YK, LFR, ESGS and B-JHvdB drafted the manuscript. TD, HM, SS, RD, AE, ACR, PWGE, MB and LV critically revised the manuscript. All authors provided final approval of the version to be submitted and agreed to be accountable for all aspects of the work in ensuring that questions related to the accuracy or integrity of any part of the work are appropriately investigated and resolved.

**Funding** DC is supported by a ZonMW grant (project number: 10430022010002).

**Competing interests** NSN and LFR are cofounders of Lipid Tools. ESGS reports personal fees from Amgen, personal fees from Sanofi-Regeneron, personal fees from Esperion, grants from Athera, outside the submitted work.

**Patient consent for publication** Not required.

**Provenance and peer review** Not commissioned; externally peer reviewed.

**Data availability statement** Data are available on reasonable request. All data relevant to the study are included in the article or uploaded as online supplemental information. The data that support the findings of this study are available from the corresponding author, on reasonable request.

**ORCID iDs**
Didier Collard http://orcid.org/0000-0001-8572-8751
Nick S Nurmohamed http://orcid.org/0000-0001-9045-6009
Yannick Kaiser http://orcid.org/0000-0001-6869-0042
Laurens F Reeskamp http://orcid.org/0000-0002-8043-951X
Liffert Vogt http://orcid.org/0000-0002-4585-7505
Bert-Jan H van den Born http://orcid.org/0000-0003-0943-4393

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
