## [Reviewer comments · BMJ Open]

ARTICLE DETAILS

TITLE (PROVISIONAL)	Cardiovascular risk factors and COVID-19 outcomes in hospitalized patients: a prospective cohort study
AUTHORS	Collard, Didier; Nurmohamed, Nick; Kaiser, Yannick; Reeskamp, Laurens; Dormans, Tom; Moeniralam, Hazra; Simsek, Suat; Douma, Renee; Eerens, Annet; Reidinga, Auke; Elbers, Paul; Beudel, Martijn; Vogt, Liffert; Stroes, Erik; van den Born, Bert-Jan

VERSION 1 – REVIEW

REVIEWER	Bassam Atallah Cleveland Clinic Abu Dhabi, UAE
REVIEW RETURNED	12-Oct-2020

GENERAL COMMENTS	The authors present a well written study from a large cohort on an important and current topic as it related to COVID-19 hospitalized patients outcomes and cardiovascular disease. Comments that need to be addressed below: - Past medical history was identified by home medications instead of by any documentation of the disease state in the chart. Medication reconciliation on admission is not always done and sometimes not comprehensive, and these medications can be prescribed for other indications The authors have highlighted this major limitation.- Particular concern is with “The combined use of beta-blockers and platelet aggregation inhibitors was used as a surrogate for history of ischemic cardiac disease.” While this combination is common in ICM but not all patients on this combination are secondary prevention patients. IS it possible to identify secondary prevention patients or at least those on DAPT?- Given the known interest in the specific classes of antihypertensive meds (eg ACEI/ARB) and outcomes, a multivariate regression analysis based on specific class of antihypertensive and possibly antidiabetic meds (eg SGLT2i) could perhaps overcome the limitation of this study and to repurpose it around medication use and outcomes. A breakdown of specific classed of meds as independent predictors and outcomes would be prudent.- Introduction paragraph 2 “during the previous of outbreak Middle East respiratory syndrome” rephrase to “during the previous outbreak of the ...”- In the inclusion you included “ high suspicion based on CT- imaging of the thorax” Did these patients ever have a positive PCR during admission? If no confirmatory positive PCR for these patients could be a major flaw, was a separate analysis done for patients with and without confirmatory test?
--

	- What were the main causes of non ICU mortality? For example were these patients having thrombotic events or other causes of death - When direct cardiac injury is discussed consider including troponin levels in the discussion, any observations on troponin from this current cohort? (Consider reviewing and citing: Atallah B, Mallah SI, AbdelWareth L, AlMahmeed W, Fonarow GC. A marker of systemic inflammation or direct cardiac injury: should cardiac troponin levels be monitored in COVID-19 patients? Eur Heart J Qual Care Clin Outcomes. 2020 Jul 1;6(3):204-207. doi: 10.1093/ehjqcco/qcaa033. PMID: 32348472; PMCID: PMC7197587.)
--	--

REVIEWER	Carlos Celis University of Glasgow, UK
REVIEW RETURNED	09-Nov-2020

GENERAL COMMENTS	The manuscript entitles “Cardiovascular risk factors are independently associated with COVID-19 mortality: a prospective cohort study” addresses a highly relevant research question. The authors have done an excellent job of reporting the association between CVD risk factors and mortality from COVID in in-hospital patients, as well as the risk of being hospitalized from CCOVID. However, there a few aspects that should be addressed to improve the quality of this manuscript. Abstract – Clear and concise Key messages box – present a good summary of the main findings and limitations of the study. Introduction - it is well written, with a clear structure. Methods – Please indicate in the methods stats section what variables was used to stratify the cohort characteristic analyses presented in table 1. The authors use the word correcting – are they referring to covariates’ adjustment? If yes, then please amend this term to “adjust”. Please indicate how many models were performed in your analyses – it seems like two models. Was this performed for all outcomes? Where proportional hazard assumptions for the cos-regression models checked? Results – Table 2 includes relevant information for the minimally adjusted model. However, it will be necessary for the reader to know whether the associations described remained when the analyses were further adjusted for smoking, obesity and the use of both a beta-blocker and antiplatelet drug. This should be added to table 2 as well as table 3. Figure S1 – should be part of the main results of this study and therefore included in the main manuscript. It may be possible to present a panel graph for fig 1 where the survival plots are presented for the 3 outcomes. Why was table 3 only focus on mortality? I was expecting to see ICU-admission and mortality in table two. Please indicate in the
--

	table title the outcome used. Saying mortality is not specific enough as you have 2 mortality outcomes in the study. Please specify if this outcome refer to mortality in the first 21 days. Why did the authors decide to threaten the exposure as 0, 1 and >1 CVD risk factor? Although the results are interesting, I think it will be important for the reader to know the association between each risk factor separately and COVID outcomes. This could be included in supplementary material if space is lacking. Some of the supplementary tables will disappear after you add the minimally and most adjusted results for all 3 outcomes in Tables 2 and 3. Please amend the results accordingly to the changes suggested. Discussion – it is well written and clear. It also highlights and identifies the key strength and limitations of the study.
--	--

VERSION 1 – AUTHOR RESPONSE

Reviewer: 1 (Bassam Atallah, Cleveland Clinic Abu Dhabi, UAE)

Comments to the Author

The authors present a well written study from a large cohort on an important and current topic as it related to COVID-19 hospitalized patients outcomes and cardiovascular disease. Comments that need to be addressed below:

- Past medical history was identified by home medications instead of by any documentation of the disease state in the chart. Medication reconciliation on admission is not always done and sometimes not comprehensive, and these medications can be prescribed for other indications. The authors have highlighted this major limitation.

RE: We thank the reviewer for his comments and thorough evaluation of our manuscript.

- Particular concern is with “The combined use of beta-blockers and platelet aggregation inhibitors was used as a surrogate for history of ischemic cardiac disease.” While this combination is common in ICM but not all patients on this combination are secondary prevention patients. IS it possible to identify secondary prevention patients or at least those on DAPT?

RE: The reviewer is correct to state that not all patients with a history of ischemic cardiac disease receive the combination of a beta-blocker and platelet aggregation inhibitor. However, in the entire cohort, only 1.8% of patients were receiving DAPT, while 10.2% of patients were using the combination of beta-blockers and antiplatelet therapy. The low prevalence of DAPT is probably explained by the fact that the guidelines recommend stopping DAPT 1 year after myocardial infarction. Therefore, we did not use the use of DAPT as a surrogate for history of ischemic cardiac disease for our analysis.

We agree with the reviewer that it is possible that not all patients on this combination are secondary prevention patients. However, the percentage of patients receiving this combination was comparable to the percentage of patients with ischemic cardiac disease in the subset of participants for whom we had more clinical data available.

- Given the known interest in the specific classes of antihypertensive meds (eg ACEI/ARB) and outcomes, a multivariate regression analysis based on specific class of antihypertensive and possibly antidiabetic meds (eg SGLT2i) could perhaps overcome the limitation of this study and to repurpose it around medication use and outcomes. A breakdown of specific classed of meds as independent predictors and outcomes would be prudent.

RE: The primary goal of the current study was to assess the relationship between cardiovascular risk factors and COVID-19 outcomes. Hence, we used the home medication list as a proxy for the

presence of CVD risk factors. For the same reason, it was difficult to assess the relationship between specific classes of antihypertensive/glucose lowering medication and COVID outcomes in our observational cohort, because we were unable to adjust for indication bias for these specific types of medication. In the discussion section (p.11), we stated: 'However, recent studies have shown no association between the use of RAS-medication and the disease course of COVID-19.(23,24)'. We think these cohorts were better suited to answering this question. For SGLT2i we are awaiting larger trials; in our cohort, the percentage of patients using SGLT2i was too low (0.25%) to perform an adequate analysis. We have added this limitation to p.12 of the discussion: '*The current study was not designed to assess the relationship between specific drugs and COVID-19 outcomes.*'

- Introduction paragraph 2 "during the previous of outbreak Middle East respiratory syndrome" rephrase to "during the previous outbreak of the ..."

RE: This error has been corrected.

- In the inclusion you included " high suspicion based on CT-imaging of the thorax" Did these patients ever have a positive PCR during admission? If no confirmatory positive PCR for these patients could be a major flaw, was a separate analysis done for patients with and without confirmatory test?

RE: Only patients with CO-RADS category 4 and 5 (respectively high and very high suspicion for COVID-19) were included in the current cohort. Previously published data shows that the CO-RADS has an excellent AUC for positive PCR values (0.91; 95%CI 0.85-0.91) (Prokop M, Van Everdingen W, Van Rees Vellinga T. CO-RADS: A Categorical CT Assessment Scheme for Patients Suspected of Having COVID-19 Definition and Evaluation. Radiology.). We clarified this in the method sections. Indeed, the majority of patients did have a positive PCR during admission (95.7%). Due to this very high percentage, and the excellent test properties of the CT assessment, we did not perform a separate analysis given the low number of patients with negative or unequivocal test results.

- What were the main causes of non ICU mortality? For example were these patients having thrombotic events or other causes of death

RE: The main cause of non-ICU related mortality was respiratory insufficiency as a result of progressive COVID-19. This also included those patients that were discharged for palliative care who, in most cases, had progressive respiratory insufficiency, but did not want or were not eligible for treatment at the ICU.

- When direct cardiac injury is discussed consider including troponin levels in the discussion, any observations on troponin from this current cohort? (Consider reviewing and citing: Atallah B, Mallah SI, AbdelWareth L, AlMahmeed W, Fonarow GC. A marker of systemic inflammation or direct cardiac injury: should cardiac troponin levels be monitored in COVID-19 patients? Eur Heart J Qual Care Clin Outcomes. 2020 Jul 1;6(3):204-207. doi: 10.1093/ehjqcco/qcaa033. PMID: 32348472; PMCID: PMC7197587.)

RE: It is a very interesting suggestion to assess the relationship between troponin and CVD risk factors in COVID. Unfortunately, troponin levels were not measured in the current cohort, hence, we did not add a section on troponin to the discussion. We have clarified this in the discussion with addition of this interesting reference.

Reviewer: 2 (Carlos Celis, University of Glasgow, UK)

Comments to the Author

The manuscript entitles "Cardiovascular risk factors are independently associated with COVID-19 mortality: a prospective cohort study" addresses a highly relevant research question. The authors have done an excellent job of reporting the association between CVD risk factors and mortality from COVID in in-hospital patients, as well as the risk of being hospitalized from CCOVID. However, there are a few aspects that should be addressed to improve the quality of this manuscript.

Abstract – Clear and concise

Key messages box – present a good summary of the main findings and limitations of the study.

Introduction - it is well written, with a clear structure.

RE: We would like to thank the reviewer for his kind comments and careful evaluation of our manuscript.

Methods –

Please indicate in the methods stats section what variables was used to stratify the cohort characteristic analyses presented in table 1.

RE: Patients were categorized based on the presence of medication based cardiovascular risk factors (hypertension, diabetes, dyslipidemia). We have clarified this in the methods section:

Patients were categorized based on the presence of either 0, 1 or more than 1 medication based cardiovascular risk factor (hypertension, dyslipidemia, diabetes).

The authors use the word correcting – are they referring to covariates' adjustment? If yes, then please amend this term to “adjust”.

RE: We are indeed referring to covariates' adjustment. We have corrected this throughout the manuscript.

Please indicate how many models were performed in your analyses – it seems like two models. Was this performed for all outcomes?

RE: We previously only reported these two models for our primary outcome, but have now added the two models for all our analyses, one correcting for age and sex, and one for other covariates (obesity, smoking and history of ischemic cardiac disease).

Where proportional hazard assumptions for the cos-regression models checked?

RE: The proportional hazard assumptions were verified using the Schoenfeld residuals. We have added this to the statistical analysis section, page 7.

Results –

Table 2 includes relevant information for the minimally adjusted model. However, it will be necessary for the reader to know whether the associations described remained when the analyses were further adjusted for smoking, obesity and the use of both a beta-blocker and antiplatelet drug. This should be added to table 2 as well as table 3.

RE: This was previously only described in the supplemental material; we have now added this to tables 2 and 3.

Figure S1 – should be part of the main results of this study and therefore included in the main manuscript. It may be possible to present a panel graph for fig 1 where the survival plots are presented for the 3 outcomes.

RE: Figure S1 is now presented in the main paper (Figure 1C).

Why was table 3 only focus on mortality? I was expecting to see ICU-admission and mortality in table two. Please indicate in the table title the outcome used. Saying mortality is not specific enough as you have 2 mortality outcomes in the study. Please specify if this outcome refer to mortality in the first 21 days.

RE: We have clarified in the legend of table 3 that we are referring to 21-day mortality, which was defined as either hospital mortality or discharge for palliative care (the primary outcome).

Why did the authors decide to threaten the exposure as 0, 1 and >1 CVD risk factor? Although the results are interesting, I think it will be important for the reader to know the association between each risk factor separately and COVID outcomes. This could be included in supplementary material if space is lacking.

RE: As suggested by the reviewer, we have performed the analysis for the effect of the specific risk factor and ICU-admission and ICU-mortality, in addition to the performed analysis with the primary outcome which can be found in table 3.

Effect of antihypertensive, lipid-lowering, and antidiabetic medications on ICU-admission				
	HR	95%CI		p
bploweringQ1	1.04	0.76	1.42	0.824

bploweringQ>1	1.13	0.82	1.55	0.461
lipidloweringQ>0	0.92	0.70	1.20	0.540
glucoseloweringQ1	1.03	0.69	1.54	0.871
glucoseloweringQ>1	1.42	1.01	2.00	0.042
Effect of antihypertensive, lipid-lowering, and antidiabetic medications on ICU-mortality				
	HR	95%CI		p
bploweringQ1	1.20	0.69	2.09	0.518
bploweringQ>1	1.29	0.74	2.25	0.360
lipidloweringQ>0	1.05	0.66	1.67	0.845
glucoseloweringQ1	1.21	0.54	2.70	0.636
glucoseloweringQ>1	2.85	1.68	4.82	0.000

We decided to present the results for mortality in our manuscript and not ICU admission, since - as stated in the discussion on p.12 - ICU-admission is difficult to interpret in the current cohort, because not all patients are admitted to the ICU as a result of advancing age or significant co-morbidities.

Some of the supplementary tables will disappear after you add the minimally and most adjusted results for all 3 outcomes in Tables 2 and 3.

Please amend the results accordingly to the changes suggested.

Discussion – it is well written and clear. It also highlights and identifies the key strength and limitations of the study.

RE: We thank the reviewer for his valuable comments and have incorporated his suggestions.

VERSION 2 – REVIEW

REVIEWER	Bassam Atallah Cleveland Clinic Abu Dhabi United Arab Emirates
REVIEW RETURNED	03-Dec-2020
GENERAL COMMENTS	The authors have adequately responded to all comments. Thank you.